# MIXTURE OF NEURAL OPERATORS: INCORPORATING HISTORICAL INFORMATION FOR LONGER ROLLOUTS

**Harris Abdul Majid & Francesco Tudisco**
The University of Edinburgh
Edinburgh, United Kingdom
`{h.abdulmajid,f.tudisco}@ed.ac.uk`

## ABSTRACT

Traditional numerical solvers for time-dependent partial differential equations (PDEs) notoriously require high computational resources and necessitate recomputation when faced with new problem parameters. In recent years, neural surrogates have shown great potential to overcome these limitations. However, it has been paradoxically observed that incorporating historical information into neural surrogates worsens their rollout performance. Drawing inspiration from multistep methods that use historical information from previous steps to obtain higher-order accuracy, we introduce the Mixture of Neural Operators (MoNO) framework; a collection of neural operators, each dedicated to processing information from a distinct previous step. We validate MoNO on the Kuramoto-Sivashinsky equation, demonstrating enhanced accuracy and stability of longer rollouts, greatly outperforming neural operators that discard historical information.

## 1 INTRODUCTION

Traditional numerical methods, such as finite difference, finite element, and spectral methods, are the standard go-to methods for solving partial differential equations (PDEs) (Thomas, 2013); however, these methods may require prohibitive computational costs, especially for high-dimensional time-dependent problems. Furthermore, traditional methods may struggle with complex geometries, and suffer from accuracy and stability issues when dealing with multiscale or nonlinear phenomena (Tadmor, 2012). Given these challenges, there has been a growing interest in the use of machine learning for solving PDEs by directly learning the solution operator (Thuerey et al., 2021; Brunton & Kutz, 2023). This paradigm shift is driven by the ability of machine learning to learn from data, identify patterns, and enable fast predictions, offering new perspectives in scientific computing.

The use of machine learning for predicting PDE trajectories can be broadly divided into two main categories. The first category involves using initial conditions to predict an entire trajectory up to a pre-specified time step (Raissi et al., 2019; Kovachki et al., 2021); however, this approach often struggles with generalizing beyond the training data. The second category adopts a sequential strategy (Stachenfeld et al., 2021; Brandstetter et al., 2022b), akin to traditional numerical methods. Here, the solution at the current time step is used to predict the solution at the next time step, which is then used as a stepping stone to make further predictions or *unroll* the trajectory. While this approach has been quite successful at predicting accurate and stable trajectories over extended periods, it is not without shortcomings. A key challenge is the accumulation of small one-step prediction errors, which compound over time, leading to significant deviations from the true trajectory. The one-step prediction errors are exacerbated at time steps that extend beyond the training range, highlighting an area of improvement in the application of autoregressive or rollout methods.

Intuition suggests that incorporating historical information in the training data, i.e. solutions at several previous time steps, will reduce one-step prediction errors and long-term predictions as a consequence, akin to traditional multistep approaches in scientific computing. However, current approaches tend to show the opposite behaviour: while incorporating historical information does reduce one-step prediction errors, it paradoxically worsens the accuracy and stability of long-term predictions (Lippe et al., 2023). As one typically trains the network to learn residuals (the difference between the solution at the next time step and the current time step), the problem arising when incor-

porating history is that the difference between the inputs is highly correlated with the model's target, the residual at the next time step. This leads the neural operator to focus on modeling second-order differences, deteriorating rollout performance.

Inspired by linear multistep methods in numerical analysis (Wanner & Hairer, 1996), which use a combination of solution values at multiple time steps, in this work we propose to use a combination of neural operators, each dedicated to processing the solution from a distinct previous time step. This collection of operators is coupled with a gating network that acts as a dynamic selector, adapting the contribution of each neural operator based on the solution at the current time step. The resulting model is an end-to-end, highly parallelizable approach that leverages historical information to, not only reduce one-step prediction errors but also, improve the accuracy and stability of long-term predictions, largely outperforming standard single-step neural operator baselines.

## 2  PROBLEM SETTING

In this paper, we focus on time-dependent PDEs of the form

$$\mathbf{u}_t + \mathcal{N}(t, \mathbf{x}, \mathbf{u}, \mathbf{u_x}, \mathbf{u_{xx}}, \dots) = 0,$$

where $t \in [0, T]$ represents the temporal dimension, $\mathbf{x} \in \mathcal{X}$ represents the (possibly multiple) spatial dimension(s), and $\mathbf{u}(t, \mathbf{x}) : [0, T] \times \mathcal{X} \to \mathbb{R}^n$ represents the solution. Here, $\mathcal{N}$ is a nonlinear operator that governs the dynamics of the system, describing how the different variables and their derivatives interact. Further, we focus on initial conditions given by $\mathbf{u}(0, \mathbf{x}) = \mathbf{u}_0(\mathbf{x})$, along with periodic boundary conditions. Discretizing the temporal dimension with step size $\Delta t$ transforms the continuous PDE into a discrete PDE, yielding a sequence of solutions at discrete time steps $\mathbf{U}_0, \mathbf{U}_1, \dots, \mathbf{U}_N$, where $N = T/\Delta t$ is the number of steps. This discretization introduces an evolution operator $\mathcal{G}$, which maps the solution at any given time step to the solution at the subsequent time step:

$$\mathcal{G}(\mathbf{U}_n) = \mathbf{U}_{n+1}.$$

In the context of operator learning (Lu et al., 2019; 2021; Li et al., 2020; Kovachki et al., 2021), the objective is to approximate the evolution operator $\mathcal{G}$ with a learned neural operator $\mathcal{G}_\theta$ that captures the dynamics of the system. A key consideration is the choice between directly predicting the solution at subsequent time steps, or predicting the differences between the solutions at two successive time steps:

$$\mathcal{G}(\mathbf{U}_n) = \mathbf{U}_{n+1} - \mathbf{U}_n.$$

Empirical observations suggest that, for *small* step sizes, predicting residuals is advantageous for maintaining accuracy and stability over extended periods (Li et al., 2021; Lippe et al., 2023). See Appendix A for further details.

In both traditional numerical methods and data-driven methods, the choice of step size $\Delta t$ is a key hyperparameter with its set of trade-offs (de Hoop et al., 2022). While a larger temporal step size decreases computational costs, it leads to larger one-step prediction errors and greater divergence from the true trajectory. In autoregressive neural methods, a larger temporal step size can significantly lower computational costs; for example, a doubling of the temporal step size halves the number of discretization points, thereby halving training and inference costs. However, if the chosen step size $\Delta t$ is too small, the trajectory can suffer from an accumulation of rounding errors.

Incorporating historical information into data-driven models is intuitively appealing, as it potentially provides the models with additional context of the underlying dynamics, thereby improving the accuracy and stability of long-term predictions. The most straightforward strategy for incorporating historical information into data-driven methods involves the introduction of additional channels, where each $m$-dimensional solution at a previous time step $\mathbf{U}_{n-1}, \mathbf{U}_{n-2}, \dots$ is assigned a separate channel, analogous to the RGB channels in color images. Using this representation as input, and mirroring the findings by Lippe et al. (2023), we observe a reduction of one-step prediction errors but a worsened high-correlation time (see Figure 1). These effects are exacerbated with smaller step sizes and longer histories. Lippe et al. (2023) explain that the problem arises because the difference between the current and previous solution $\mathbf{U}_n - \mathbf{U}_{n-1}$ is highly correlated with the residual of the next time step $\mathbf{U}_{n+1} - \mathbf{U}_n$—this leads the learned neural operator to focus on second-order differences which is known to deteriorate performance in explicit schemes.

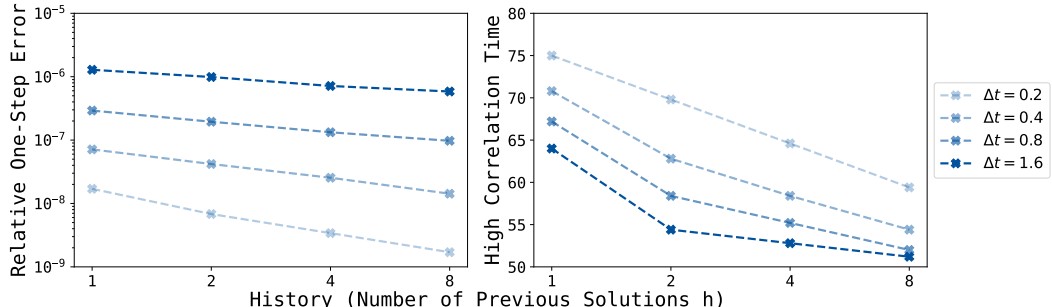

Figure 1: Performance of a neural operator (UNet) evaluated on Kuramoto-Sivashinsky data with step sizes $\Delta t = \{0.2, 0.4, 0.8, 1.6\}$. **Left:** Relative One-Step Error as a function of History. **Right:** Time taken for correlation to drop below a threshold of 0.9 as a function of History.

To mitigate the aforementioned challenges, we propose training a collection of neural operators, each dedicated to processing the solution from a distinct previous time step, before weighting and aggregating their outputs. This framework prevents the model from focusing on higher-order differences, allowing effective use of historical information.

## 3  MIXTURE OF NEURAL OPERATORS

Akin to traditional numerical methods where lower-order accurate methods can be extended to obtain higher-order accuracy by using multiple time steps, in this work we introduce a novel ensemble approach, termed Mixture of Neural Operators (MoNO), which synergizes a collection of neural operator models $\mathcal{G}_\theta^{(0)}, \ldots, \mathcal{G}_\theta^{(h-1)}$, orchestrated by a gating network $R_\theta$. The MoNO framework is designed to dynamically weigh and integrate the predictions from each operator. Formally,

$$\text{MoNO}_\theta\left(\mathbf{U}_n, \ldots, \mathbf{U}_{n-h+1}\right) := \sum_{i=0}^{h-1} R_\theta\left(\mathbf{U}_n, \ldots, \mathbf{U}_{n-h+1}\right)_i \cdot \mathcal{G}_\theta^{(i)}(\mathbf{U}_{n-i}),$$

where $R_\theta(z)_i$ denotes the gating network's weighting for the $i$-th operator, and $\mathcal{G}_\theta^{(i)}(z)$ denotes the corresponding neural operator's output.

The training objective for MoNO optimizes a loss function $\mathcal{L}(\theta)$ that minimizes the discrepancy between the aggregated prediction and the true residual:

$$\mathcal{L}(\theta) = \left|\text{MoNO}_\theta\left(\mathbf{U}_n, \ldots, \mathbf{U}_{n-h+1}\right) - \left(\mathbf{U}_{n+1} - \mathbf{U}_n\right)\right|.$$

This design encourages a cooperative interaction among the neural operators, where adjustments in one operator's parameters induce compensatory adaptations across the ensemble, ensuring a cohesive evolution towards minimizing residual prediction errors.

To realize the MoNO framework, we explore various neural operator architectures, including UNet and Fourier Neural Operator (FNO), allowing for a diverse representation of the solution mappings. The flexibility of MoNO extends to the configuration of each neural operator, enabling adjustments in parameter allocation, activation functions, and architectural nuances to tailor the ensemble's predictive capacity. Central to the efficacy of MoNO is the gating network, $R_\theta$, which determines the proportional contribution of each neural operator to the ensemble prediction. We use a softmax linear layer to ensure that the experts' weights are non-negative and sum to one, facilitating interpretability and maintaining balanced contributions.

Furthermore, MoNO's framework is inherently amenable to distributed computing environments. Leveraging standard model parallelism techniques, the framework can distribute its components across multiple GPUs, routing inputs to their respective neural operators for processing and subsequently aggregating the outputs, thus capitalizing on computational resources efficiently.

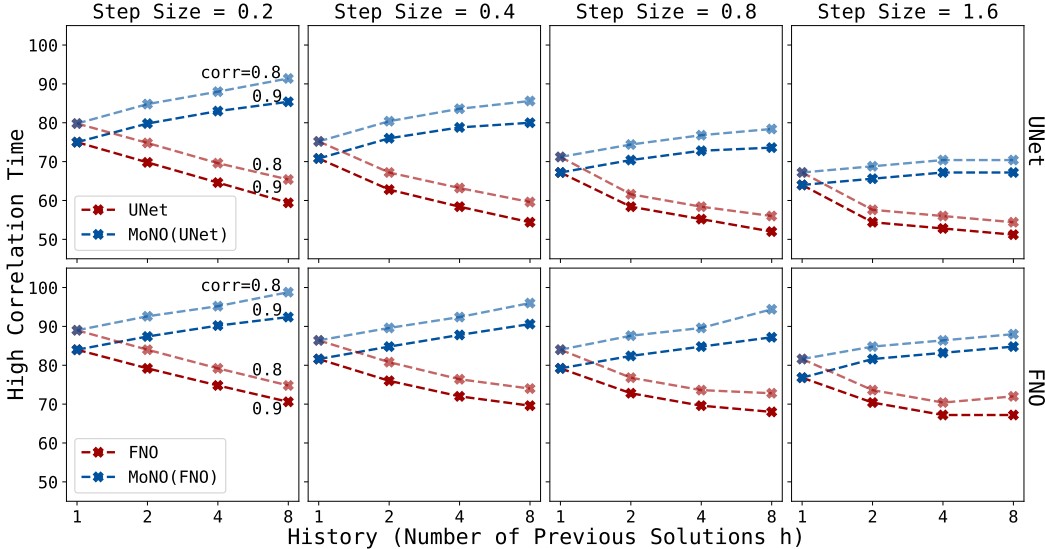

Figure 2: High-Correlation Time of MoNO (and baselines) evaluated on Kuramoto-Sivashinsky data. Each graph reports the High-Correlation Time as a function of History, and each column corresponds to a different step size. **Top:** Comparison between UNet and MoNO(UNet). **Bottom:** Comparison between FNO and MoNO(FNO).

## 4 EXPERIMENTS

In order to test the performance of the proposed model we present experimental evaluation on the one-dimensional Kuramoto-Sivashinsky (KS) equation. The KS equation is a fourth-order nonlinear PDE derived to model diffusive-thermal instabilities in laminar flame fronts (Kuramoto, 1978; Sivashinsky, 1977). Its one-dimensional variant can be expressed as:

$$u_t + u_{xx} + u_{xxxx} + uu_x = 0.$$

Here, the fourth-order derivative $u_{xxxx}$ and the nonlinear term $uu_x$ contribute to complex and chaotic behavior which present a challenge for traditional numerical solvers (Hyman & Nicolaenko, 1986; Kevrekidis et al., 1990; Smyrlis & Papageorgiou, 1991), necessitating fine spatial and temporal discretizations (and therefore increased computational costs) for accurate and stable trajectories.

We assessed the performance of MoNO using KS data across various step sizes and history lengths. The objective was to predict the residual: the difference between successive time steps, $\mathbf{U}_{n+1} - \mathbf{U}_n$. As a baseline neural operator $\mathcal{G}_\theta$ we used the U-Net by Gupta & Brandstetter (2022), which is frequently used for neural PDE solvers Lippe et al. (2023). A softmax-MLP serves as MoNO's gating network. We conducted parallel evaluations using another prevalent model, the Fourier Neural Operator (FNO) with 8 Fourier layers and 32 Fourier modes (Li et al., 2020). To ensure equitable comparisons, we adjusted the channels of each model so that the model has approximately 50 million trainable parameters. For training, we chose the Mean Squared Error (MSE) as a loss metric. To evaluate, we autoregressively apply the models on the initial conditions in the testing dataset, and report the high-correlation time: the duration for the Pearson correlation between the true trajectory and predicted trajectory to fall below thresholds of 0.8 and 0.9.

At a step size of 0.2, the U-Net baseline obtained a high-correlation time of 75.0 (79.8) for a threshold of 0.9 (0.8). An increase in history length up to 8 steps resulted in reduced high-correlation times, manifesting as 59.4 (65.4) for a threshold of 0.9 (0.8); whereas, the MoNO framework demonstrated significant improvements, recording 14%, 28%, and 44% longer high-correlation times at history lengths of 2, 4, and 8, respectively. This trend persisted across all step sizes. It is noteworthy that at step size $\Delta t = 1.6$, extending the history from 4 to 8 did not yield further improvements, suggesting that data from 4 steps prior (or 6.4 seconds) lacked relevance for subsequent predictions. Similar to U-Net, MoNO with FNO backbone consistently outperformed the baseline across all history lengths and step sizes. See Figure 2, top (U-Net) and bottom (FNO) panels.

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

## A  DIRECT VERSUS RESIDUAL PREDICTION

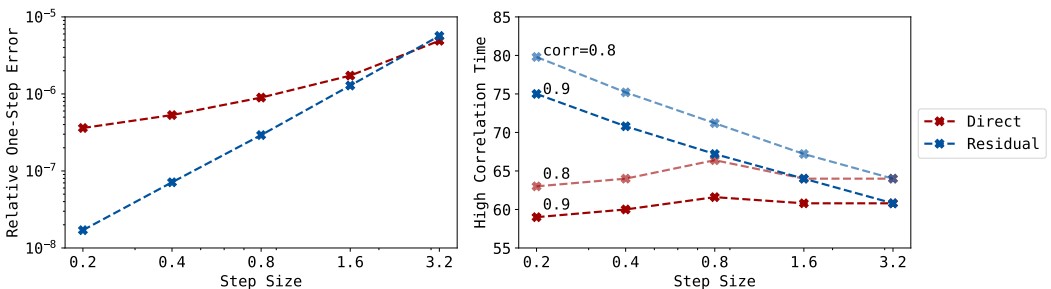

Figure 3: Comparing direct and residual prediction. Direct prediction achieves higher relative one-step error and lower high-correlation time up to $\sim 2$ seconds as observed by Li et al. (2021) and Lippe et al. (2023).

## B  DATA GENERATION

For our experiments, we generate data for the one-dimensional Kuramoto-Sivashinsky equation according to Brandstetter et al. (2022a)[1], who use the method of lines, with the spatial derivatives computed using the pseudo-spectral method. We set $T = 100$ for the training (and $T = 200$ for validation and testing) data, with $\Delta t = 0.2$. Further, we set $\mathcal{X} = [0, 64]$, with $\Delta x = 0.25$. Initial conditions were sampled from a distribution over the truncated Fourier series with random coefficients $A_k \sim U(-0.5, 0.5)$, $l_k \sim \{1, 2, 3\}$, and $\phi_k \sim U(0, 2\pi)$:

$$u_0(x) = \sum_{k=1}^{10} A_k \sin\left(\frac{2\pi l_k x}{L} + \phi_k\right),$$

where $L = 64$ is the length of the spatial domain. Finally, we set periodic boundary conditions.

We generated 2048 trajectories for the training dataset and 128 trajectories for each of the validation and testing datasets. For each trajectory, the first 360 steps were considered to be part of the *warmup phase* and subsequently discarded. The data was initially generated using double-precision floating-point format (`float64`) and then converted into single-precision floating-point format (`float32`) for our experiments.

---

[1]We make slight modifications to their code (`https://github.com/brandstetter-johannes/LPSDA`) and set $\Delta t = 0.2$.

