# OpenReview forum: "Mixture of Neural Operators:  Incorporating Historical Information for Longer Rollouts"
_ICLR.cc/2024/Workshop/AI4DiffEqtnsInSci — AI4DiffEqtnsInSci @ ICLR 2024 Poster_

### Official Review · Reviewer_rNYe · 2024-02-26
**Solid evaluation of a decent method that explicitly considers multiple history frames in timeseries forecasting.**

**Rating:** 7
**Confidence:** 4

**Review:**

### Summary
A Mixture of Neural Operators (MoNO) is introduced that trains multiple Neural Operators, each of them learning to predict a target frame $x_{t+1}$ on basis of a particular frame from the past. A gating network decides on the weighting of each Neural Operator's contribution to the final solution. This approach allows the explicit incorporation of multiple historic frames to base a prediction of the next frame.

### Strenghts
- Appealing idea that tackles a relevant problem in temporal dynamics forecasting.
- Clear problem statement and paper organization.
- Proper experimentation supports the claims.

### Weaknesses
- Training/inference time and memory trade-off between single vs multi NODE model would be informative to assess the applicability of the introduced method.
- The statement "This leads the neural operator to focus on modeling second-order deteriorating rollout performance" needs some verification (referencing prior work or experimental results).
- Can you show some empirical results to underline your statement "This design encourages a cooperative interaction among the neural operators, where adjustments in operator’s parameters induce compensatory adaptations across the ensemble, ensuring a cohesive towards minimizing residual prediction errors."? It would be great to see what actually happens in the background, i.e., how $R_{\theta}$ adjusts its weights, or whether a single Neural Operator basically dominates all contributions to a prediction.

### Questions
- How would you expect your method to compare to multi-step forecasting methods, where a single model is trained to predict multiple forecast horizons, such as described in **Dynamics Forecasting** paragraph in Section 4 of [[1]](https://arxiv.org/abs/2306.01984)?
- Have you tried to play with the inputs of the gating network $R_{\theta}$? I'm wondering, whether $R_{\theta}$ would benefit from receiving the respective outputs of the individual Neural Operators.

### Minor Comments
- Around the middle of page two, a reference seems to be missing ("See ?? for further details.").

### References
[1] https://arxiv.org/abs/2306.01984

---

### Official Review · Reviewer_ABiq · 2024-02-27
**A good paper demonstrating novel framework for learning NO for leveraging historical information.**

**Rating:** 6
**Confidence:** 4

**Review:**

This paper proposes a novel framework for learning neural operators that can leverage historical information from previous time steps to improve the accuracy and stability of long-term predictions for time-dependent partial differential equations (PDEs). The framework, called Mixture of Neural Operators (MoNO), consists of an ensemble of neural operators, each dedicated to processing the solution from a distinct previous time step, and a gating network that dynamically weighs and aggregates their outputs. The paper demonstrates the effectiveness of MoNO on the Kuramoto-Sivashinsky equation, a challenging nonlinear PDE that exhibits complex and chaotic behavior.

## Pros
1. The paper introduces a new and innovative approach for incorporating historical information into neural operators, inspired by traditional multistep methods in numerical analysis. The paper also provides a clear explanation of the motivation and intuition behind the proposed framework, as well as the challenges and limitations of existing methods.
2. Performance: The paper shows that MoNO outperforms standard single-step neural operator baselines, such as U-Net and Fourier Neural Operator, across various step sizes and history lengths, in terms of reducing one-step prediction errors and increasing high-correlation time.

## Cons
1. The paper only evaluates MoNO on one PDE, the Kuramoto-Sivashinsky equation, which limits the generalizability and applicability of the proposed framework. The paper could benefit from testing MoNO on other PDEs, especially those with higher dimensions, different boundary conditions, or different types of nonlinearities, to demonstrate its robustness and versatility.
2. The paper only compares MoNO with standard single-step neural operator baselines, which may not be the most relevant or competitive methods for incorporating historical information. The paper could benefit from comparing MoNO with other data-driven methods that use historical information, such as recurrent neural networks, attention mechanisms, or memory networks, to provide a more comprehensive and fair evaluation.
3. The paper could benefit from providing more insights and explanations of why and how MoNO works, such as the role of the gating network, the effect of the history length, the trade-off between accuracy and stability, or the potential sources of error and instability. The paper could also provide more quantitative and qualitative measures of the performance and the behavior of MoNO, such as error bounds, convergence rates, sensitivity analysis, or error propagation.

---

### Meta-Review · Area_Chair_x8zT · 2024-02-28

**Recommendation:** Accept (Poster)

**Metareview:**

Both reviewers appreciate the novelty of the work and vote for acceptance. The authors propose to use further historical information within FNO to act as a temporal multi-step method from numerical analysis. It is nice to bring concepts from numerical analysis into machine learning and see them succeed. I also vote for acceptance but agree with Reviewer ABiq that the method should be compared to other methods that use the same amount of historical information rather than the prior step.

---

### Decision · Program_Chairs · 2024-02-28

Accept (Poster)